Effect of lag screw on stability of first metatarsophalangeal joint arthrodesis with medial plate

http://orcid.org/0000-0002-0522-7752 Daszkiewicz Karol 1 kardaszk@pg.edu.pl
Rucka Magdalena 1
Czuraj Krzysztof 2
http://orcid.org/0000-0002-2458-0703 Andrzejewska Angela 1
Łuczkiewicz Piotr 2 3
1 Department of Mechanics of Materials and Structures, Faculty of Civil and Environmental Engineering, Gdańsk University of Technology , Gdańsk , Poland
2 Pomeranian Reumatology Center , Sopot , Poland
3 Second Clinic of Orthopaedics and Kinetic Organ Traumatology, Medical University of Gdansk , Gdańsk , Poland
Williams John
Electronic publication date: 2024 Feb 28
Publication date: 2024
Volume: 12
Electronic Location ID: e16901
Received 2023 Jul 5; Accepted 2024 Jan 17
Copyright: © 2024 Daszkiewicz et al.
Copyright year: 2024
Copyright holder: Daszkiewicz et al.
License: This is an open access article distributed under the terms of the Creative Commons Attribution License, which permits unrestricted use, distribution, reproduction and adaptation in any medium and for any purpose provided that it is properly attributed. For attribution, the original author(s), title, publication source (PeerJ) and either DOI or URL of the article must be cited.
License URL: https://creativecommons.org/licenses/by/4.0/

Keywords: MTP-1 joint, Arthrodesis biomechanics, Lag screw, Medial plate, Interfragmentary movement, Great toe

Funding: National Science Center of Poland 2021/05/X/ST8/00374 This work was supported by the National Science Center of Poland [grant number 2021/05/X/ST8/00374]. The funders had no role in study design, data collection and analysis, decision to publish, or preparation of the manuscript.

==============================
Background

First metatarsophalangeal joint (MTP-1) arthrodesis is a commonly performed procedure in the treatment of disorders of the great toe. Since the incidence of revision after MTP-1 joint arthrodesis is not insignificant, a medial approach with a medially positioned locking plate has been proposed as a new technique. The aim of the study was to investigate the effect of the application of a lag screw on the stability and strength of first metatarsophalangeal joint arthrodesis with medial plate.

Methods

The bending tests in a testing machine were performed for models of the first metatarsal bone and the proximal phalanx printed on a 3D printer from polylactide material. The bones were joined using the locking titanium plate and six locking screws. The specimens were divided into three groups of seven each: medial plate and no lag screw, medial plate with a lag screw, dorsal plate with a lag screw. The tests were carried out quasi-static until the samples failure.

Results

The addition of the lag screw to the medial plate significantly increased flexural stiffness (41.45 N/mm vs 23.84 N/mm, p = 0.002), which was lower than that of the dorsal plate with a lag screw (81.29 N/mm, p < 0.001). The similar maximum force greater than 700 N (p > 0.50) and the relative bone displacements lower than 0.5 mm for a force of 50 N were obtained for all fixation techniques.

Conclusions

The lag screw significantly increased the shear stiffness in particular and reduced relative transverse displacements to the level that should not delay the healing process for the full load of the MTP-1 joint arthrodesis with the medial plate. It is recommended to use the locking screws with a larger cross-sectional area of the head to minimize rotation of the medial plate relative to the metatarsal bone.

Introduction

First metatarsophalangeal (MTP-1) joint arthrodesis is a commonly performed procedure in the treatment of disorders of the great toe (Gaudin et al., 2018). A good outcome following this procedure depends on two factors: mechanical stability of the bone during the postoperative period and sufficient blood supply to the bone (Claes, 2021). The excessive transverse and opening displacements between the bones may delay or even suppress the bone formation process (Augat et al., 2003; Claes et al., 2018). Therefore, the relative displacements between the bones and the stiffness of fixation systems are used as measures of the mechanical stability of the bone arthrodesis (Claes, 2021). The successful bone union without hardware failure and bone fracture is necessary to achieve pain relief and functional improvement. The maximum load allows to assess the risk of failure of fixation system or bone fracture. The current works focus on new minimally invasive techniques of arthrodesis stabilisation strong enough to allow weight bearing without loss of the initial stiffness of the bone fixation (Fuld et al., 2019; Hodel, Viehöfer & Wirth, 2020). Since the incidence of surgical revision after MTP-1 joint arthrodesis is not insignificant (Gaudin et al., 2018), a medial approach with a medially positioned locking plate was proposed. This technique offers e.g., protection of the vascular supply to the metatarsal head, excellent exposure, superior cosmetic result, safe distance from the dorsomedial cutaneous nerve and reinforcement of medial hallux correction in hallux valgus deformity (Dayton, Glynn & LoPiccolo, 2001). A pilot clinical trial showed that MTP-1 joint arthrodesis is safe procedure and give clinical results comparable to those obtained with currently widely used techniques (Kuik & Łuczkiewicz, 2021).

Significant reduction of interfragmentary motion (Märdian et al., 2019) and faster fracture healing (Horn et al., 2011) was demonstrated when an interfragmentary lag screw was used in a locking plate construct. The addition of a lag screw to the dorsal plate improves significantly the stability and strength of the bone fusion in the MTP-1 joint (Buranosky et al., 2001; Politi et al., 2003; Campbell et al., 2017). Relative transverse displacements between bones close to 1.5 mm were reported for the medial approach in a study (Witkowski et al., 2021), which may adversely affect the healing process (Augat et al., 2003). We propose the use of lag screw as a method to minimize such deformation in the medial approach. So far there is no published study that quantifies the biomechanical performance of MTP-1 joint arthrodesis with a medially positioned locking plate and lag screw. Therefore, the purpose of the study was to investigate the effect of the lag screw on the stiffness and strength of the MTP-1 joint arthrodesis with the medial plate.

Numerical and experimental results from a previous study (Witkowski et al., 2021) have suggested that the medial plate may provide better stability of the MTP-1 joint arthrodesis than a dorsally positioned locking plate due to the greater vertical bending stiffness of the medial plate. However, only one sample was tested for each technique and the results were compared to the weak dorsal plate construct in Witkowski et al. (2021). Therefore, the secondary objective was to perform experimental tests on a larger number of samples and compare the results between the medial plate constructs and the dorsal plate with lag screw, described as the most stable fixation method (Politi et al., 2003; Harris, Moroney & Tourné, 2017).

Materials and Methods

Specimen preparation

Geometry of the metatarsal bone and proximal phalanx was taken from the freely accessible anatomic model of a right foot (Giovinco, 2013). The specimen preparation process in Hypermesh software (Altair Engineering, Troy, MI, USA) is shown in Fig. 1. The STL geometry was scaled in Hypermesh software to obtain the proximal phalanx length consistent with the mean value of 32.7 mm reported in (Munuera, Polo & Rebollo, 2008). Then the bone surfaces in the MTP-1 joint were virtually cut to a congruent spherical shape, simulating the use of a concave proximal reamer and a convex distal reamer during surgery. The dorsal and medial surfaces of bones were also trimmed to facilitate proper fixation of the plate. A part with a rectangular cross-section was added at the “distal” end of the proximal phalanx to enable specimens to be loaded at the centre of pressure under the great toe. The metatarsal bone was cut and a cylindrical part was added to allow the bone to be fixed in a jig as close to the MTP-1 joint as possible. The geometry of the bones prepared in this way was used in the 3D printing of composite joint models.

Figure 1 The process of bone samples preparation for arthrodesis in Hypermesh software.

Artificial models of the metatarsal bone and proximal phalanx were prepared with fused filament fabrication (FFF), using an Ultimaker 3 Extended 3D printer (Ultimaker B.V., Utrecht, Netherlands). In the manufacturing process of bone models, the Ultimaker white polylactide was applied. Due to the irregular shape of bone models, the water-soluble Ultimaker polyvinyl alcohol was used to build the supports. The selected parameters of the manufacturing process are described in Table 1.

Table 1 Values of parameters used in fused filament fabrication of artificial bone models.

Parameter	Bone model	Support	
Filament type	Polylactide	Polyvinyl alcohol	
Nozzle diameter	0.4 mm	0.4 mm	
Primary layer height	0.1 mm	0.1 mm	
Top solid layers	4	1	
Bottom solid layers	4	1	
Outline/perimeter shells	3	3	
Internal fill pattern	Lines	Lines	
Interior fill percentage	100%	100%	
Extruder temperature	200 °C	215 °C	
Heated bed temperature	60 °C	60 °C	
Default printing speed	70 mm/s	35 mm/s	

Three techniques were used in the MTP-1 joint arthrodesis (see Fig. 2): medially positioned locking plate (Medial LP);

dorsally positioned locking plate with interfragmentary lag screw (dorsal LP lag screw);

medially positioned locking plate with interfragmentary lag screw (medial LP lag screw).

Figure 2 Tested fixation constructs in dorsal, plantar, medial and lateral views.

(A) Medial LP. (B) Dorsal LP lag screw. (C) Medial LP lag screw.

The arthrodesis of the seven pairs of bones for each group was performed by an experienced orthopaedic surgeon using the same titanium locking plate (short metatarsophalangeal plate, Medgal Sp. z o.o, Księżno, Poland) and six 2.4 mm locking screws. The right plate was used in the dorsal position and the left plate in the medial position, as it was better suited to the geometry of the proximal phalanx in the medial approach. In the second and third groups, an additional 2.7 mm cortical lag screw was used between the first metatarsal and proximal phalanx. The relative position of bones during arthrodesis was standardized using a dedicated composite jig (Fig. 3).

Figure 3 Dedicated composite jig for the MTP-1 joint arthrodesis.

Biomechanical testing

A total of 21 samples were tested in a Zwick/Roell Z10 universal testing machine (ZwickRoell GmbH & Co. KG, Ulm, Germany). The cylindrical end of the first metatarsal and the rectangular end of the proximal phalanx were clamped in custom-made stainless steel jigs, see Fig. 4A. Such an experimental setup did not limit the relative displacements between bones and simulated the worst loading conditions with the fixed metatarsal bone and the proximal phalanx loaded perpendicular to the plantar surface of the great toe. An initial load of 1 N was applied in the tests. The MTP-1 joint samples were loaded to failure with a constant speed of 5 mm/min of the upper crosshead to simulate quasi-static loading conditions during the bending test.

Figure 4 The experimental setup during the bending test.

(A) Overall view. (B) View from the DIC camera with marking the analysis points (P1, P2, P3).

To characterize the mechanical behaviour of the MTP-1 joint during the bending test, the digital image correlation (DIC) method was utilized. Images of the lateral surface of the samples covered with a stochastic pattern were captured every 1 s by two cameras with a resolution 4096 × 3000 pixels, being a part of an ARAMIS MC 3D 12M system (GOM GmbH, Braunschweig, Germany). After measurements, the collected photographs were processed by ARAMIS Professional software. Figure 4B shows the view from the left camera at the beginning of the test. To analyse the displacements during the bending test, three control points were selected (P1, P2 and P3, see Fig. 4B).

Data analysis

A construct flexural stiffness was calculated by fitting the least squares regression line to a force-displacement curve over the 5–200 N range. Additionally, an initial flexural stiffness was computed as the initial slope of this curve. The vertical displacement at point P3 was used in the stiffness calculations because point P3 corresponds to the centre of pressure under the great toe. The 200 N load was chosen as an estimate of the maximum force under the great toe which may be 25% of the body weight in the stance phase of gait (Ledoux & Hillstrom, 2002). The relationship between the force (5–200 N range) and vertical relative displacement between points P1 and P2 was used to determine the shear stiffness of the fixation.

Maximum load to the first failure was defined as the extreme load until a force drop greater than 50 N was observed. The horizontal and vertical displacements between points P1 and P2 for two levels of load (50 N and 200 N) were calculated as a measure of the opening displacement and transverse displacement at the joint space, respectively. The force of 50 N corresponds to a partial load of the MTP-1 joint in a healing shoe in the postoperative period.

Statistical analysis

An analysis of variance (ANOVA) was performed using R version 4.2.1 software (R Core Team, 2022) to examine the effect of the fixation technique on the stiffness of the MTP-1 joint arthrodesis and the maximum load. The homogeneity of variance was verified using Levene’s test (centre = median). Tukey’s Honest Significant Difference (Tukey’s HSD) method with a 95% family-wise confidence level was employed for the pair-wise comparisons between the fixation constructs. The interfragmentary displacements and the shear stiffness were compared between all pairs of fixation techniques using Tamhane’s T2-test for unequal variances.

Results

The addition of an interfragmentary lag screw to the medially positioned plate increased the initial flexural stiffness by 153% and the shear stiffness by 352%, see Table 2. The dorsal plate with the lag screw was the stiffest construct (Figs. 5A, 5B). Significant differences in stiffness were observed (p < 0.05, p-values are given in the Supplemental File) between all pairs of fixation techniques used in the MTP-1 joint arthrodesis. A greater initial flexural stiffness than the flexural stiffness in the range of 5–200 N was observed especially in the constructs with the interfragmentary lag screw. The greatest mean maximum load was obtained for the medially positioned plate (Table 2), but the difference between this group and the dorsal LP lag screw and the medial LP lag screw groups was not significant (p = 0.70 and p = 0.52, respectively). Results for all samples from the Zwick/Roell machine are available in the dataset (Daszkiewicz et al., 2022a).

Table 2 Comparison of mean values of flexural stiffness, initial flexural stiffness, shear stiffness and maximum load, and their standard deviations (SDs) for different fixation methods.

Fixation method	Medial LP	Dorsal LP lag screw	Medial LP lag screw	
Flexural stiffness N/mm (SD)	23.84 (6.67)	81.29 (10.7)	41.45 (6.3)	
Initial flexural stiffness N/mm (SD)	25.37 (6.98)	107.30 (13.19)	64.10 (9.56)	
Shear stiffness N/mm (SD)	128.24 (63.92)	2469.96 (640.7)	579.47 (151.96)	
Maximum load N (SD)	850.77 (234.18)	773.70 (136.38)	744.86 (144.72)	

Figure 5 Comparison of mean values and standard deviations obtained for different fixation methods.

(A) Flexural stiffness (N/mm). (B) Shear stiffness (N/mm). (C) Relative displacements (mm) between bones for a force of 50 N. (D) Relative displacements (mm) between bones for a force of 200 N. *adjusted p-value < 0.05.

The greatest relative displacements between bones for the two levels of load were observed for the medial LP group (Table 3, Figs. 5C, 5D). The differences between this group and the dorsal LP lag screw and the medial LP lag screw groups were significant (p < 0.05, p-values are given in Supplemental File). A significant decrease of 88% and 80% in the mean transverse displacement was obtained for the medial LP lag screw group compared to the group without the lag screw for the force of 50 and 200 N, respectively. The mean opening displacements were smaller for the dorsal LP lag screw group than for the medial LP lag screw group (Table 3), but these differences were not statistically significant (Figs. 5C, 5D). The mean transverse displacement was almost four times smaller than the mean opening displacement in the dorsal LP lag screw group, while 80% greater transverse displacements than the opening displacements were observed for the medial LP specimens for a force of 200 N (Table 3). The displacements of points P1–P3 for the full range of load are available in the dataset (Daszkiewicz et al., 2022b).

Table 3 Comparison of mean values of relative displacements between bones and their standard deviations (SDs).

Group	Force	Medial LP	Dorsal LP lag screw	Medial LP lag screw	
Opening displacement mm (SD)	50 N	0.256 (0.099)	0.045 (0.010)	0.065 (0.035)	
200 N	0.976 (0.185)	0.317 (0.113)	0.484 (0.180)	
Transverse displacement mm (SD)	50 N	0.429 (0.140)	0.014 (0.003)	0.051 (0.032)	
200 N	1.761 (0.592)	0.084 (0.020)	0.347 (0.148)	
Note:

The displacements were computed for a force of 50 and 200 N.

The shear fracture of one or two locking screws inserted into the metatarsal bone (Fig. 6A) was observed between the screw head and shank in the majority of the specimens with the medial plate (Fig. 6F). The medial LP with lag screw specimens typically failed due to a fracture of the proximal phalanx in the plane of the interfragmentary lag screw insertion (Fig. 6B) preceded by the fracture of the locking screw (Fig. 6F). Medial LP and dorsal LP lag screw constructs also failed by fracture of the proximal phalanx in the plane of the most distal locking screw insertion (Fig. 6C). Fracture of the metatarsal bone in the plane of the locking screws insertion was observed only in the medial LP group (Fig. 6D). The constructs with the interfragmentary lag screw failed also due to a shear fracture of the metatarsal bone at the steel jig (Fig. 6E).

Figure 6 Failure modes observed in experimental tests and comparison of maximum loads.

(A) Shear fracture of locking screws inserted into the metatarsal bone. (B) Fracture of the proximal phalanx in the plane of the lag screw insertion. (C) Fracture of the proximal phalanx at the insertion of the most distal locking screw. (D) Fracture of the metatarsal in the plane of the locking screws insertion. (E) Shear fracture of the metatarsal at the fixation site. (F) Maximum loads (N) for different fixation methods. Letters indicate failure mode for each sample. N/A–the failure mode was not identified.

Discussion

The previous studies (Politi et al., 2003; Harris, Moroney & Tourné, 2017) shown that the combination of the dorsally positioned plate with interfragmentary lag screw provides the greatest stiffness of the bone fusion in the MTP-1 joint. This was confirmed by our results, as significantly lower stiffness was obtained for the fixation methods with the medially positioned plate. The dorsal LP lag screw construct was more than 10 times more stable than the dorsal plate only, considering the opening displacements (Politi et al., 2003; Campbell et al., 2017). In the case of the medially positioned plate, the lag screw increased the flexural stiffness by 74% and two times reduced the mean opening displacement for a force of 200 N. The lower effect of the lag screw on the flexural stiffness in the medial approach is due to that the lag screw and the locking plate work as a beam web in this approach, while they form flanges of composite I-beam structure in the dorsal approach. Consequently, the medial plate with lag screw did not provide better stability than the dorsal plate with lag screw, as in the case of constructs without lag screw (Witkowski et al., 2021). However, several times greater influence of the lag screw on the shear stiffness was observed, which resulted in the reduction of the transverse relative displacements by 80–88%.

The stability of the fixation construct has a significant influence on the quality of new bone and number of molecules related to blood vessel formation in the healing area (Lienau et al., 2005, 2010; Claes, 2021). Tensile and shear strains in the callus area resulted in four times lower vascularization and two times lower bone formation compared to compression strain (Claes et al., 2018; Claes & Meyers, 2020). As the experimental measurements of strains are difficult, we reported the values of opening and transverse displacements between the metatarsal and proximal phalanx bones (Table 3) which correspond to the state of tension and shear, respectively. The initial axial interfragmentary movements (IFMs) of 0.2–1 mm led in the diaphyseal bone to good bone healing (Goodship & Kenwright, 1985; Schell et al., 2008; Claes, 2021), while a value of 2 mm is usually assumed as a critical value above which bone formation process may be delayed or even suppressed (Lowenberg, Nork & Abruzzo, 2008; Campbell et al., 2017). The values of relative bone displacements for all fixation constructs were lower than 0.5 mm for the load of 50 N. Consequently, they should not delay the healing process for the partial weight bearing of the MTP-1 joint. However, very small relative displacements of less than 0.1 mm obtained for the constructs with the lag screw may delay bone healing underneath the locking plate, because too stiff fixation may suppress the stimulation of new bone formation (Hente et al., 2004; Bottlang et al., 2010; Röderer et al., 2014). On the other hand, stiffer fixation is advantageous for smaller gaps used in MTP-1 joint arthrodesis, because the tissue strain in the callus increases with decreasing gap size (Perren & Cordey, 1977).

The high union rate after MTP-1 joint arthrodesis with full immediate postoperative weight-bearing was observed for different fixation methods (Berlet, Hyer & Glover, 2008; Abben, Sorensen & Waverly, 2018; Patel et al., 2019). Those results showed that early weight-bearing and return to regular activity may be a safe postoperative protocol. Therefore, we have presented the relative displacements for a force of 200 N as an approximation of the maximum force in the MTP-1 joint for full weight-bearing (Gefen et al., 2000; Ledoux & Hillstrom, 2002). The mean opening displacements for this force are in the optimal range of 0.2–1 mm for all fixation techniques. However, the mean transverse displacement for the medial locking plate was greater than the critical initial shear movement of 1.5 mm which delayed the healing process in an animal study (Augat et al., 2003). Moreover, the mean shear stiffness of 128.2 N/mm for this fixation method is substantially lower than the critical value of 380 N/mm above which optimal conditions for bone healing were found in (Epari et al., 2007) for a 1 mm fracture gap. The addition of the lag screw to the medial plate significantly increased the shear stiffness to 579.5 N/mm and reduced the mean transverse displacement to 0.347 mm for a force of 200 N. Consequently, the application of the lag screw in the medial approach could be particularly beneficial in case of immediate full weight-bearing of the MTP-1 joint after surgery.

There were no significant differences in strength between fixation techniques. The mean maximum load was greater than the values of 350–400 N given in a study (Harris, Moroney & Tourné, 2017) in which the force was applied on a larger lever arm and about four times greater than the estimated critical force of 200 N. The fracture of the proximal phalanx in the plane of the interfragmentary lag screw insertion (Fig. 6B) was more often observed for the medial plate than for the dorsal plate (Fig. 6F). This suggests that the lag screw should be inserted more dorsally through the MTP-1 joint, but this was difficult due to the position of the locking screws for the titanium plate used. Failure at the interfragmentary lag screw insertion plane was typically preceded by the fracture of the most plantar locking screw, which increased the load carried by the lag screw. The fracture of the proximal phalanx was observed also in the plane of the most distal locking screw insertion (Fig. 6C), due to the small cross-sectional area of the phalangeal shaft. Such a mode of failure was observed for all dorsal LP lag screw constructs in the study (Harris, Moroney & Tourné, 2017).

The shear fracture of one or two proximal locking screws inserted into the metatarsal bone (Fig. 6A) was observed in 8/14 specimens with the medially positioned plate due to a combination of bending moment and large shear vertical force at the interface between the screw head and shank. Such damage of the locking screws was not observed for the dorsally positioned plate in the current study and literature (Harris, Moroney & Tourné, 2017), because axial forces act instead of shear forces in the locking screws in the dorsal approach. The locking screws with hexagon socket S2.5 were used in the current study to simplify the modelling of the head geometry in subsequent numerical studies. However, the relatively large size of the hexagon socket compared to the diameter of the screw head resulted in a small cross-sectional area at the head-shank junction, which made the screws susceptible to plasticity and shear failure. The fracture of the locking reduces the stability of MTP-1 joint arthrodesis and may cause migration of the screw head. In the pilot clinical study (Kuik & Łuczkiewicz, 2021) locking screws with a larger diameter of 2.8 mm were used and only one patient experienced locking screw migration. The shear fracture was not observed in the previous study (Witkowski et al., 2021), where the locking screws with torx socket T8 and greater cross-sectional area were used. Consequently, the use the locking screws with a larger cross-sectional area of the head is recommended for the medial plate constructs.

Observation of the damaged medial plate specimens indicated that the IFMs between bones were mainly caused by the rotation of the plate relative to the locking screw in the oblong hole at the metatarsal bone. Consequently, despite greater flexural stiffness of the titanium plate in the medial approach (Witkowski et al., 2021) significantly lower stiffness was obtained for the medial plate constructs than for the dorsal LP lag screw group. The plate rotation was the effect of plastic deformation of the bone and the screw shank on the two locking screws closest to the MTP-1 joint (Fig. 6A). Therefore, plates dedicated to the medial approach should have a geometry and screw configuration that minimizes plate rotation. The greatest stiffness in the medial LP group was obtained for the sample in which strong contact between the bones was observed during loading at the lateral side of the joint, hence the introduction of compression between the bones is particularly important for the constructs without the lag screw.

Our study has some limitations. The influence of muscle forces on deformation of the MTP-1 joint was omitted, but the greatest relative displacements between bones are mainly the effect of the ground reaction force. The results obtained for the artificial bone models may not accurately reflect the mechanical behaviour of real bones. However, the shear modulus of synthetic bones of 817 MPa (Sabik et al., 2022) is comparable to the equivalent shear modulus of 1,047 MPa for the real human first metatarsal bone (Danesi et al., 2012; Coşkun, Çelik & Kişioğlu, 2023). The use of own artificial bone models allowed to virtually prepare samples for arthrodesis and thus minimize the standard deviation compared to real bones and alternatives like sawbones. Moreover, the influence of different mechanical properties of cadaveric samples and the lack of repeatability of their fixation in the steel jigs were eliminated. As a result, statistically significant results were obtained for seven samples per group. Only quasi-static loading was applied to the specimen during the experiments. Cycling loading may reduce the stiffness and the maximum load due to fatigue failure. The experimental tests were performed for one type of the locking plate and one geometry of the MTP-1 joint. Different geometries of the locking plate and bones will be analysed in further research.

Conclusions

The addition of the lag screw to the medially positioned plate significantly increased the shear stiffness in particular and reduced relative displacement between bones for the full load of the MTP-1 joint to the level that should not delay the healing process. Large maximum loads and low values of relative displacement obtained for a force of 50 N suggested that all fixation methods are safe for the partial load of the joint. However, the constructs with the medially positioned plate were less stable compared to the dorsal plate combined with interfragmentary lag screw. It is recommended to use the locking screws with a larger cross-sectional area of the head and plates with a such screw configuration that minimizes the rotation of the plate relative to the metatarsal bone in the medial approach. Further clinical trials are necessary to confirm conclusions obtained with the artificial bone models.

Supplemental Information

Supplemental Information 1 The adjusted p-values obtained for pairwise comparisons of initial flexural stiffness, flexural stiffness (Tukey’s HSD method) and shear stiffness (Tamhane’s T2-test).

Supplemental Information 2 The adjusted p-values obtained in Tamhane’s T2-test for pairwise comparisons of opening and transverse relative displacements between different fixation methods.

Supplemental Information 3 Values of flexural stiffness, initial stiffness, shear stiffness and maximum load for all specimens.

Raw data used in the statistical analysis.

Supplemental Information 4 Relative opening and transverse displacements between bones for all specimens.

Raw data for a force of 50 N and 200 N used in the statistical analysis.

Additional Information and Declarations

Competing Interests

Author Contributions

Data Availability

The authors declare that they have no competing interests.

Karol Daszkiewicz conceived and designed the experiments, performed the experiments, analyzed the data, prepared figures and/or tables, authored or reviewed drafts of the article, and approved the final draft.

Magdalena Rucka performed the experiments, analyzed the data, prepared figures and/or tables, authored or reviewed drafts of the article, and approved the final draft.

Krzysztof Czuraj performed the experiments, authored or reviewed drafts of the article, and approved the final draft.

Angela Andrzejewska conceived and designed the experiments, performed the experiments, prepared figures and/or tables, authored or reviewed drafts of the article, and approved the final draft.

Piotr Łuczkiewicz conceived and designed the experiments, authored or reviewed drafts of the article, and approved the final draft.

The following information was supplied regarding data availability:

The data is available at the university repository MOST WIEDZY:

- Bending test results from Zwick/Roell Z10 universal testing machine: Daszkiewicz, K., Rucka, M., Andrzejewska, A. J., Czuraj, K., & Łuczkiewicz, P. (2022). Bending test results of first metatarsophalangeal joint after arthrodesis with medially or dorsally positioned locking plate and lag screw. [dataset]. Gdańsk University of Technology. https://doi.org/10.34808/a33s-pj64

- Displacements of bones during bending test captured by ARAMIS MC 3D 12M system: Daszkiewicz, K., Rucka, M., Andrzejewska, A. J., Czuraj, K., & Łuczkiewicz, P. (2022). Displacements of bones during bending test of first metatarsophalangeal joint after arthrodesis with medially or dorsally positioned locking plate and lag screw. [dataset]. Gdańsk University of Technology. https://doi.org/10.34808/wqxn-8z65.

The flexural stiffness, initial stiffness, shear stiffness, maximum load and relative displacements between bones for all specimens are available in Supplemental Files.

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
