# Peer review of "Effect of lag screw on stability of first metatarsophalangeal joint arthrodesis with medial plate"

_PeerJ, doi:10.7717/peerj.16901_

## Round 0.1 · original submission · Major Revisions

Two reviewers have recommended that you revise and resubmit your manuscript. One of the issues that was raised by the reviewers is the need to provide some discussion for the clinical motivation and justification of the proposed method of arthrodesis. The choice of substitute materials rather than cadaver bones should be addressed. Are you able to compare some of the basic biomechanical properties of your printed bone substitutes to the bones themselves? You state that the stiffness of the synthetic bone is comparable to the stiffness of real bones (lines 271-272) but do not provide a reference. What about the fracture properties? Without such information the wording in the discussion and conclusion should be more cautionary and should include a statement of limitations requiring further work with cadaver tissue. This may be especially relevant to the data in Table 5 and Figure 5 and to statements regarding "safe levels of loading (line 38)?

In your introduction lines 51-53 you state that prior results have suggested that a medial plate may provide better stability than a dorsal plate. Shouldn't you circle back to this question in your discussion and conclusions?

I also recommend modifying the text where the word 'fused' is used. I find this confusing. Perhaps you could describe this as a fixation construct to achieve arthrodesis or bone fusion. In the Materials and Methods section, line 79 please insert the word 'virtually': "Then the bone surfaces in the MTP-1 joint were 'virtually' cut to ..." In the Data Analysis section, line 125 and in Table 2 please add 'flexural' between initial and stiffness.


·

Basic reporting

Good standard of English language. Satisfactory background, methodology and analysis of results.

Experimental design

Good design and hypothesis. Use of 3d printed models is a limiting factor, but mentioned by authors. However, cadaveric specimens are also limited. Good presentation of results.

Validity of the findings

Findings justified, however authors tempted to suggest medial plating with an interfragmentary screw is a safe option despite results suggesting dorsal plating with inter fragmentary screw was superior.

·

Basic reporting

This article is clearly written, using professional English throughout.
The Literature references are sufficient. Data are shared and the figures and tables are comprehensive

Experimental design

The primary aim of the study is well stated. However from a clinical/patient perspective you could question whether the experimental design is meaningful and/or relevant.
In current clinical practise the use of a plate and lag screw has been widely introduced. The use of a medial plate is from patient perspective not desirable.
In addition the indication for this procedure is of mucht more importance compared to the fixation type.

Validity of the findings

The novelty is limited.
All underlying date are robust but of less adede value. The research techniques are robust and statistically sound. However the implications for clinical use are limited, due to the model and the patient perspective.

Additional comments

Authors should consider the effect of this fixation type in relation to currently widley used techniques. Inclusion of a patient study group would be very important, inclusion of patient perspective is essential.

Reviewer 3 ·

Basic reporting

Please refer the reviewer comments document.

Experimental design

Please refer the reviewer comments document.

Validity of the findings

Please refer the reviewer comments document.

Additional comments

Please refer the reviewer comments document.

Annotated reviews are not available for download in order to protect the identity of reviewers who chose to remain anonymous.

---

## Round 0.2 · accepted · Accept

I should request that you correct the text for minor language issues. Please change "In specific" to "Specifically" in lines 100, 228, and 262.